# Interplay between disorder and electronic correlations in compositionally complex alloys

David Redka [1,2], Saleem Ayaz Khan[1], Edoardo Martino [3], Xavier Mettan [3], Luka Ciric[3], Davor Tolj [3], Trpimir Ivšić [3,9], Andreas Held[4], Marco Caputo[5], Eduardo Bonini Guedes [5], Vladimir N. Strocov [5], Igor Di Marco [6,7], Hubert Ebert[4], Heinz P. Huber [1,2] ✉, J. Hugo Dil [3,5], László Forró[3,8] & Ján Minár [1] ✉

Owing to their exceptional mechanical, electronic, and phononic transport properties, compositionally complex alloys, including high-entropy alloys, represent an important class of materials. However, the interplay between chemical disorder and electronic correlations, and its influence on electronic structure-derived properties, remains largely unexplored. This is addressed for the archetypal CrMnFeCoNi alloy using resonant and valence band photoemission spectroscopy, electrical resistivity, and optical conductivity measurements, complemented by linear response calculations based on density functional theory. Utilizing dynamical mean-field theory, correlation signatures and damping in the spectra are identified, highlighting the significance of many-body effects, particularly in states distant from the Fermi edge. Electronic transport remains dominated by disorder and potentially short-range order, especially at low temperatures, while visible-spectrum optical conductivity and high-temperature transport are influenced by short quasiparticle lifetimes. These findings improve our understanding of element-specific electronic correlations in compositionally complex alloys and facilitate the development of advanced materials with tailored electronic properties.

Compositionally complex alloys (CCAs), which comprise the diverse range of medium- to high-entropy alloys (HEAs), are an exciting class of materials, consisting of randomly distributed multi-principal elements on crystalline lattices[1,2]. Aiming on HEAs, the denomination stems from the entropy term overruling the enthalpy of formation of individual phases when mixing a large number of elements, hence escaping phase separation. They exhibit exceptional mechanical attributes including elevated toughness, minimal plastic deformation, and enhanced tensile and yield strengths[3,4], along with intriguing electronic as well phononic transport properties[5–8]. CCAs, including HEAs, bridge the structural gap between crystalline solids and amorphous materials, exhibiting long-range periodicity but with atom

[1]New Technologies Research Center, University of West Bohemia, Plzen, Czech Republic. [2]Department of Applied Sciences and Mechatronics, Munich University of Applied Sciences HM, Munich, Germany. [3]Institute of Physics, École Polytechnique Fédérale de Lausanne, Lausanne, Switzerland. [4]Department of Chemistry, Ludwig-Maximilians-University Munich, Munich, Germany. [5]Photon Science Division, Paul Scherrer Institut, Villigen, Switzerland. [6]Institute of Physics, Nicolaus Copernicus University, Toruń, Poland. [7]Department of Physics and Astronomy, Uppsala University, Uppsala, Sweden. [8]Stavropoulos Center for Complex Quantum Matter, Department of Physics and Astronomy, University of Notre Dame, Notre Dame, IN, USA. [9]Present address: Department of Physical Chemistry, Ruđer Bošković Institute, Zagreb, Croatia. ✉e-mail: heinz.huber@hm.edu; jminar@ntc.zcu.cz

variations on lattice sites inducing site disorder akin to Anderson localization[9]. For years, the question of electron propagation in such an environment has persisted[10], lacking the translational invariance ensuring the validity of the Bloch theorem for propagating electronic waves, yielding electronic localization. Empirical observations report electric resistivity ($\rho$) approaching the Ioffe-Regel limit with a subdued $d\rho/dT$ dependence[11], demonstrating that the effect of increasing residual resistivity in Cantor-Wu alloys may be linked to magnetic disorder effects[12]. Regarding the $d\rho/dT$ dependence various explanations have been suggested, including Anderson localization[13], or quantum interference in accordance with Mooij correlations[14]. For the latter, it becomes evident that, in addition to electron-phonon and spin scattering, many-body effects arising from the electron-electron interaction may play a significant role. However, to date, this issue has not been thoroughly investigated.

In this work, we probe the role of electronic correlation next to disorder effects in CrMnFeCoNi, likely the most studied HEA and prototype CCA, by employing resonant (ResPES) and valence-band (VB) photoemission spectroscopy (PES), optical conductivity and temperature dependent electrical resistivity measurements. All these measurements are supported and explained by electronic structure calculations based on density functional theory (DFT) and dynamical mean-field theory (DMFT). The results demonstrate that chemical and magnetic disorder in CCAs predominantly influence the electronic properties in the vicinity of the Fermi level. In contrast, electronic many-body effects gain significance when probing electronic structure-derived properties involving states distant from the Fermi edge. This is exemplified by the case of electrical conductivity at high temperatures and optical properties in the visible and UV spectral range.

## Results and discussion

### Element resolved photoemission spectroscopy

In ResPES employing X-ray absorption at the $L_3$-edge, photon energies proximate to the edge excite photoelectrons in the direct photoemission channel ($2p^6 3d^n + \hbar\omega \rightarrow 2p^6 3d^{n-1} + e_f$) and in the channel of the dipole transition of a core electron to an unoccupied state ($2p^6 3d^n + \hbar\omega \rightarrow 2p^5 3d^{n+1}$). At the $L$-edges the second channel typically dominates (at the $M$-edges the two channels have comparable magnitudes and interfere, giving rise to the resonant Fano profile) and the subsequent decay of this intermediate state ($2p^5 3d^{n+1} \rightarrow 2p^6 3d^{n-1} + e_f$), akin to an Auger process, gives rise to a strong resonant enhancement of the ResPES signal[15,16]. Upon progressive increase of photon energy above the edge, ResPES from the VB typically evolves through two regimes, depending on the core-hole and conduction-band lifetimes: (1) coherent ResPES, where the core electron excited into the conduction band is coupled with VB electron ejected to vacuum[17,18]. In this case the spectral peaks stay at constant binding energy ($E_B$) and their shape reflects the element-specific partial density of states (pDOS) of the VB; (2) incoherent resonance Auger regime, where the conduction-band electron is decoupled from the two VB electrons, one filling the core hole and another ejected to vacuum[19]. In this case the spectral peaks stay at constant kinetic energy ($E_k$), and their shape is related to the self-convolution of the pDOS. To summarize, our ResPES measurements reveal site- or element-specific information on the electronic structure suitable for probing complex alloys where band-overlapping or hybridization is commonplace[20].

Element-specific $L_3$ X-ray absorption spectroscopy (XAS) data are depicted next to ResPES photoelectron energy distribution curves (EDCs) in Fig. 1 on the binding energy ($E_B$) scale relative to the Fermi energy $E_F$ (calibrated by the Fermi level of gold).

For Cr, and in contrast to the other elements, the EDCs exhibit a pronounced maximum at constant $E_B$ of 1.8 eV for an extended range of incident photon energies. This indicates that the coherent ResPES contribution (constant $E_B$) dominates over the resonant Auger one (constant $E_k$) in the whole shown photon-energy range close to the $L_3$ XAS maximum[20-23]. For pure Cr, a VB peak in the EDCs of ResPES measurements was observed at 1.2 eV[23]. For Mn we find an EDC maximum at the $L_3$ edge at 3.6 eV, with a faint shoulder extending up to $E_B = 7.5$ eV, hinting to a small resonant Auger contribution at constant $E_k$ above the resonance. For Fe, Co, and Ni, the main features of the EDCs are clearly attributable to the Auger process[22-25] (see Supplementary Information SI for a more detailed EDC plot on the specific elements). The EDC maxima for the XAS $L_3$ edge are 4.7 eV, 4.2 eV, and 7.2 eV $E_B$, respectively. In no case a dominant VB contribution at constant $E_B$ is observed. Comparing these data with those of pure elements provides information on band filling, hybridization, electronic correlations, chemical disorder and crystal field effects. Ni, sharing the fcc crystal structure and comparable lattice constant with the CrMnFeCoNi HEA[6], mainly allows for a focus on the effects induced by chemical disorder. The measured EDCs display a shift of the well-known 6 eV satellite for pure Ni[26,27] towards 7.2 eV in the CrMnFeCoNi HEA. Shifts as large as 1.4 eV towards higher $E_B$ have been observed for Ni based alloys and intermetallic compounds[25,28,29], and explained by $d$-band filling through the hybridization of wave functions located on different lattice sites after alloying with more electropositive elements[28,30,31].

Figure 2a depicts the calculated $d$-band partial density of states (pDOS) of individual elements, for LDA (yellow solid line) and LDA + DMFT (blue solid line), next to the EDCs from Fig. 1 (black solid line). For calculation details see "Methods" and SI. Since we are interested only in the peak position, all graphs are normalized to their maximum. Marked differences emerge between LDA + DMFT and pure LDA, including strong satellites (Mn, Co, and Ni) and a generalized band-narrowing (Ni and Co). Eventually, the spectral weight (excluding satellites) shifts to lower $E_B$ by including DMFT. For Cr and Fe, $d$-band satellites merge with the $sp$-pDOS (see detailed figures on band resolved pDOS in the SI). With increasing $d$-band filling, there is a progressive split-off of the formed Hubbard bands towards higher $E_B$, although this effect is mitigated between Mn and Fe by the large difference in $U$. In Cr and Fe, the value of $U$ is so small with respect to the bandwidth that no detached satellite peak can form; rather, it reflects a renormalization of the DOS, altering the $d$-band shape from rectangular to triangular. The positions of the satellites in the pDOS are compared to the ResPES data. For Mn, the small shoulder in the ResPES data at 7.5 eV coincides with the satellite at 8 eV in the pDOS. A comparable analysis for Fe and Co is not possible. For Ni, the pDOS data reveal a satellite at 8.2 eV $E_B$, elevated by 1 eV compared to that determined experimentally via ResPES.

### Element specific on-site Coulomb interaction

We apply the Cini-Sawatzky Theory (CST)[32-34] by comparing the measured EDCs (assumed as Auger spectra) with the self-convoluted single-particle pDOS. Within the CST framework, Auger signals may be categorized into distinct regimes, according to the ratio between electronic bandwidth $W$ and on-site Coulomb interaction $U$. For $U \gg W$, the spectra correspond to the quasi-atomic limit with split-off satellites at high $E_B$, whereas being band-like for $U \ll W$. For the latter Lander[35] proposes that the Auger signal equals the self-convolution of the single-particle band. At $U \sim W$, a complex interplay occurs, resulting in the superposition of both states. According to the CST, for systems with nearly filled $d$-bands, a discernible shift of the Auger spectra towards lower $E_B$ is found, displacing the maximum of the self-convolution of the DOS by $U$ relative to the Auger signals[28,36,37]. This is explained by the energy difference of the two-hole state and the two one-hole states being equal to the Coulomb interaction on the two-particle energy scale. The ResPES maxima (black solid lines) are aligned with the self-convolution which are depicted in Fig. 2a by dashed colored lines. It is evident, that for Cr, only the ResPES contribution is measured, as the EDCs spectral shape coincides with that of the

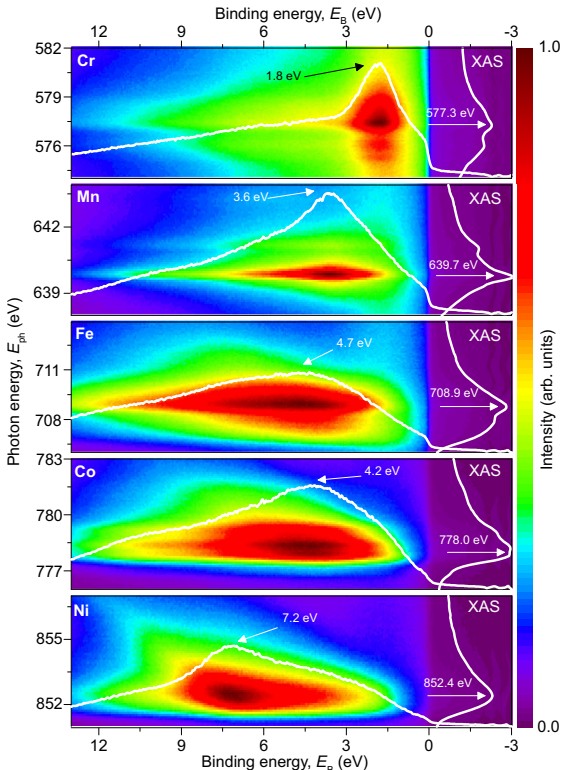

**Fig. 1 | Element-specific X-ray absorption spectroscopy (XAS) and resonant photoemission spectroscopy (ResPES) for the CrMnFeCoNi HEA.** The figure shows XAS $L_3$-edge spectra (right side of each panel, in white, with intensity in arbitrary units) and corresponding ResPES energy distribution curves (EDCs) for Cr, Mn, Fe, Co, and Ni. The EDCs are plotted on a binding energy ($E_B$) scale, with the photon energy as the vertical axis. The white lines on the ResPES plots indicate the EDC intensities at the XAS $L_3$ maxima, marked by arrows. For Cr, the EDCs exhibit a pronounced peak at a constant $E_B$, indicating the presence of valence band features. In contrast, for Fe, Co, and Ni, the EDCs display maxima at constant kinetic energy, suggesting significant Auger contributions and highlighting the transition from radiationless ResPES (constant $E_B$ below XAS maximum) to resonant Auger regimes (constant $E_k$ XAS $L_3$ maximum). The shift of the 6 eV satellite in pure Ni towards higher $E_B = 7.2$ eV in the alloy, may be attributed to the d-band filling after alloying with more electropositive elements.

*sp*-pDOS. For Mn and Fe, which are approximately situated in the bandlike limit with $U < W$, the self-convolution and ResPES data on the two-electron scale show a congruence in their peaks. For Co, where the EDC is dominated by the resonant Auger spectral weight, a minor offset of approximately 1.6 eV is identified with $U$, whose value is slightly smaller than our suggested $U$ of 2.5 eV (see "Methods"). For Ni, the corresponding offset amounts to 4 eV, which is slightly larger than the value of 3 eV used as $U$ in the LDA + DMFT calculation. This trend is expected, considering the different filling of the $3d$-band[28,38–40], as well as the limitations of the DMFT solver[41]. In order to investigate the sensitivity of the CST we calculate the CrMnFeCoNi HEA with the initially given $U$ values, but increasing them for Fe from 1.5 eV to 2 eV and Ni from 3 eV to 4 eV. The pDOS (solid line) as well as the self-convoluted signal (dashed line) are given in Fig. 2a as green lines. For Fe there is barely a difference in the pDOS visible with a slight increase of the satellite. No shift in the self-convolution signal is found. For Ni, the $d$-block shifts also barely recognizably towards the Fermi edge, but the correlation satellite splits off significantly towards higher $E_B$ of almost 10 eV. Consequently, the main peak of self-convolution hardly changes. The difference between EDC and self-convolution is 4 eV, which corresponds exactly to the Hubbard $U$ from the LDA + DMFT calculations. The variation of the element-specific $U$ value of Fe and Ni does not

change the pDOS of the other elements (see the more detailed plot in the SI). However, since the extreme displacement of the split-off satellite is not observed experimentally, we keep the initial chosen pure element $U$ values in our calculations.

Figure 2b presents VB PES measurements for $\hbar\omega = 1200$ eV, alongside one-step model calculations[42] for the LDA as well LDA + DMFT potentials (see SI for details). Experimentally, a satellite feature is visible, attributable to Ni at approximately 7 eV, as corroborated by ResPES. The LDA + DMFT calculation reveals a peak at approximately 8 eV, which perfectly overlaps with the Ni satellite in the calculated pDOS. The offset between theory and experiment is attributable to the perturbative nature of the DMFT solver, and aligns well with a 1 eV offset observed in pure Ni[43], which confirms our choice of $U$ a posteriori. Despite the satellite, the shoulder spanning roughly from 3 eV to 4 eV, paralleling the experimental observations, is also reproduced in the LDA + DMFT calculation. The plateau ranging from 5 eV $E_B$ to 8 eV in the LDA + DMFT calculation resonates well with the experimental data, which extends from 4 eV $E_B$ to 7 eV. The LDA approach fails to adequately capture any of these correlation fingerprints. Despite the strong agreement between experimental and theoretical results, we still observe a discrepancy in the bandwidth, which is slightly narrower in LDA + DMFT. This difference can however have an extrinsic origin, as e.g., arise from subtle contributions from a minor surface oxide layer (see the oxygen 1 s peak in the wide scan XAS measurement provided in the SI). Also, the experimental background (intensity at lowest $E_B$) accounts for a systematic deviation. Besides, we are able to conclude that the assumption of Hubbard $U$ parameters for CrMnFeCoNi, aligned with the pure metals, is justified, and electronic correlations play a site-specific role, comparable to that of pure $3d$ transition metals.

### Broadening of the band structure

The influence of the broadening of states due to chemical disorder and quasiparticle lifetimes can be discerned in the computed Bloch spectral function (BSF) depicted in Fig. 2c, d for LDA and LDA + DMFT, respectively. Near $E_F$, both calculations reveal a scarcely dispersive $d$-band block with strongly localized electrons, while the parabolic dispersion of the $sp$-bands becomes apparent at higher $E_B$. The $d$-bands around 2 eV are for the LDA + DMFT case so extensively smeared, especially along the symmetry line X-Γ-L, that sub-bands cannot be resolved. The strongly localized satellite states are perceptible over a constant smeared background up to 9 eV. Arguing qualitatively, the spectral width of the $d$-states implies that the lifetimes of these electrons must be exceedingly short. Interestingly, this arises mainly from correlation effects at high $E_B$, but from the combined action of correlations and chemical/magnetic disorder in the vicinity of $E_F$.

### Element resolved quasiparticle lifetimes

Quasiparticle lifetimes $\tau$ can be obtained from the self-energy function $\Sigma$ from DMFT via $\hbar/\tau = 2\text{Im}(\Sigma)$. Figure 3a displays element-specific lifetimes for Fe and Ni, obtained by evaluating the Greens function on the real energy axis. Data for other elements are plotted in the SI. For context, black lines show pure element calculations in their natural crystal structures, alongside corresponding experimental data from literature[43–46]. These calculations align well, although the Ni lifetimes for excited states above 1 eV slightly exceed experimental observations. For all elements, near $E_F$, a Fermi liquid theory-like behavior emerges[46], with $\tau \propto (E - E_F)^{-2}$. To assess the effects of altered lattice constants on the element-specific self-energy, lifetimes for Fe and Ni within the fcc lattice but the HEA's lattice constant are illustrated in yellow in Fig. 3a. While Ni shows a negligible variation, $\gamma$-Fe exhibits notably reduced lifetimes, despite

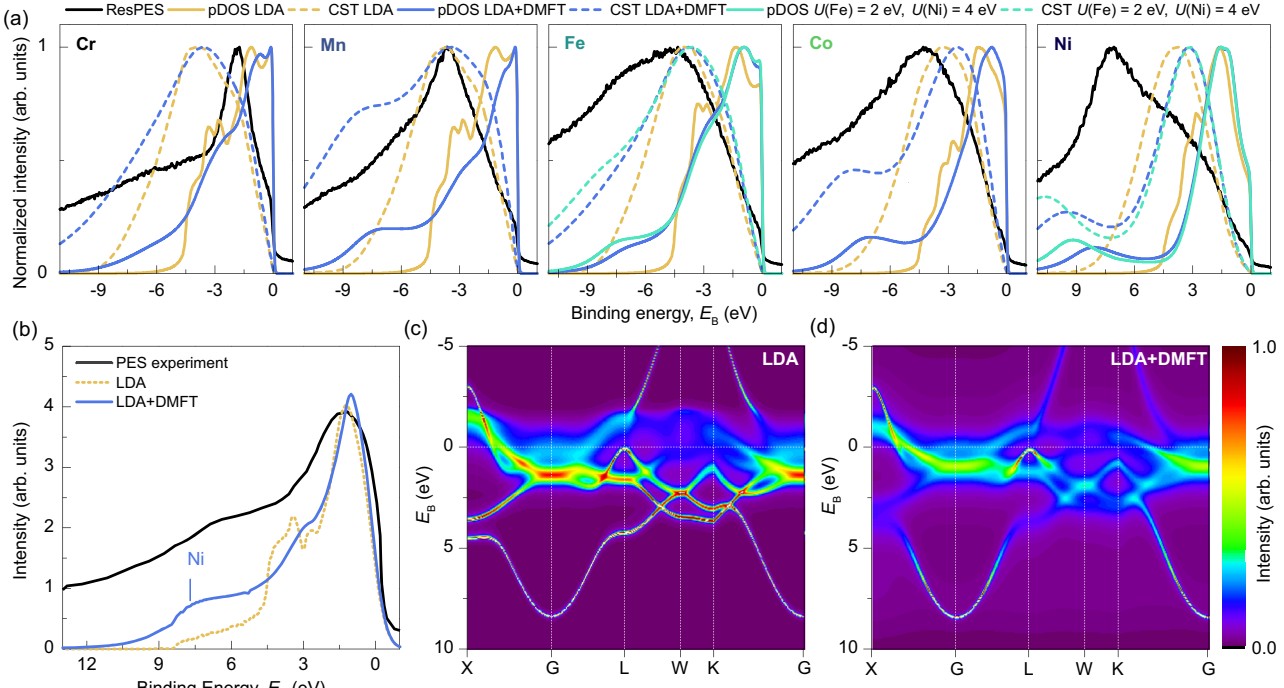

**Fig. 2 | Element-resolved electronic spectra of the CrMnFeCoNi HEA.**
**a** Calculated partial density of states (pDOS) for Cr, Mn, Fe, Co, and Ni in the CrMnFeCoNi alloy using LDA and LDA + DMFT, shown as yellow and blue solid lines, respectively. Measured energy distribution curves (EDCs) at the XAS $L_3$ absorption maximum as black solid lines. The self-convolutions of the pDOS (Cini-Sawatzky Theory, CST) are given by dashed lines in corresponding colors. For Fe and Ni, additional pDOS calculations using LDA + DMFT with modified Hubbard $U$ values are shown as green lines. In the pDOS the LDA + DMFT approach introduces satellite features not present in the pure LDA results for all elements. For Cr, the EDCs align well with the pDOS, being valence band like, while Mn shows good overlap between the EDCs and CST maxima. In contrast, Fe, Co, and Ni exhibit increasing distance between EDCs and self-convolution peaks. Lager $U$ values for Fe and Ni shift the satellites towards higher binding energies without significantly

altering the overall pDOS. **b** Comparison of experimental valence band photo-emission spectroscopy (PES) data (black line) with one-step model calculations for LDA (yellow dashed line) and LDA + DMFT (blue line). For LDA + DMFT a shoulder at 8 eV (Ni marker), which is absent in the LDA results is found, and the LDA + DMFT spectrum is more smeared overall. The offset between PES and calculations at high binding energies is due to the experimental background not mimicked in the calculations. **d** Calculated Bloch spectral function (BSF) for CrMnFeCoNi using LDA (**c**) and LDA + DMFT (**d**). The color maps represent the spectral intensity. For LDA, the states near the Fermi energy are already smeared through disorder, whereas for LDA + DMFT, bands at higher binding energies also exhibit significant smearing due to reduced quasiparticle lifetimes, to the extent that subbands may not be resolved, as seen between the Γ and L points.

having the same Hubbard $U$. This is not surprising, considering that γ-Fe is known to have stronger magnetic fluctuations leading to a complex magnetic landscape[47]. The bottom row of Fig. 3a contrasts the pure elements in the fcc HEA structure against the CPA-derived disordered paramagnetic state. Here, both elements show an increased $\tau$, akin to those in their natural crystals. For the LDA + DMFT calculation of CrMnFeCoNi with $U$ = 4 eV for Ni and 2 eV for Fe, the green lines in Fig. 3a depict the results. For Ni, lifetimes above $E_F$ demonstrate a 3/4 reduction compared to the 3 eV calculation, reflecting the ratio of $U$ values. A comparable trend is given for Fe. As observed in the pDOS data, other elements are not affected by the variation of $U$ (see SI). Our calculations indicate a minor influence of chemical disorder on DMFT derived lifetimes, particularly when contrasted with data for the pure elements and taking the effect of changing crystal structure into account. Furthermore, $\tau$ exhibit a nearly linear dependency to variations in $U$.

## Frequency resolved optical conductivity
The dynamic response of electrons in the CrMnFeCoNi HEA is further probed by means of complex optical conductivity $\sigma(\omega)$ measurements. Typically, $\sigma(\omega)$ reveals for metals a Drude peak in Re($\sigma(\omega)$), which broadens with increasing scattering rate and eventually may merge with existing higher-energy interband transitions. Thus separation into inter- and intra-band contributions becomes challenging for correlated 3d transition metals. We therefore compute the $\sigma(\omega)$ tensor via

the Kubo formalism[48] by the current-current correlation function[49].

$$\sigma_{\mu\nu}(\omega) = \frac{i\hbar}{\pi^2} \frac{1}{\Omega} \int_\Omega d^3r \int_\Omega d^3r' \int_{E_B}^\infty dE' \int_{E_B}^\infty dE \, \Theta_T(E - E_F) \Theta_T(E_F - E')$$
$$\times \left\{ \frac{\text{tr}\left[j_\mu(\mathbf{r})\text{Im}G^+(E')j_\nu(\mathbf{r}')\text{Im}G^+(E)\right]}{\left(E' - E - i\Gamma_{ep}\right)\left(\hbar\omega + E - E' + i\Gamma_{ep}\right)} + \frac{\text{tr}\left[j_\nu(\mathbf{r}')\text{Im}G^+(E')j_\mu(\mathbf{r})\text{Im}G^+(E)\right]}{\left(E' - E - i\Gamma_{ep}\right)\left(\hbar\omega + E' - E + i\Gamma_{ep}\right)} \right\}$$
$$(1)$$

Due to the fully relativistic formulation of the current density operator $j(\mathbf{r})$, both paramagnetic and diamagnetic terms are implicitly included, with the latter yielding a Drude-like contribution[49,50]. However, as in our approach electron-phonon collision driven damping mechanisms are not considered a priori, a phenomenological complex photon energy $\Gamma_{ep}$ is introduced overruling the infinite small lifetime $0^+$ from adiabatic switching on of the external perturbative field. We adopt a constant $\Gamma_{ep}$ value of 0.340 eV, attributed to the substantially diminished electronic mean free path, proximate to the lattice parameter, as well the given Fermi velocity for such disordered alloys[51]. Our calculations utilize both LDA and LDA + DMFT, with the latter including energy dependent electron-electron scattering $\Gamma_{ee}$ through the imaginary part of the self-energy inherently incorporated in the Greens function. Although the DMFT scheme solely considers $d$-band states in $\Sigma$[52], an empirical choice of $\Gamma_{ee}$ for the $sp$-band is not obligatory. It has been shown analytically[53] as well numerically within the $GW$ formalism[54] that $d$-band screening dominates the total self-

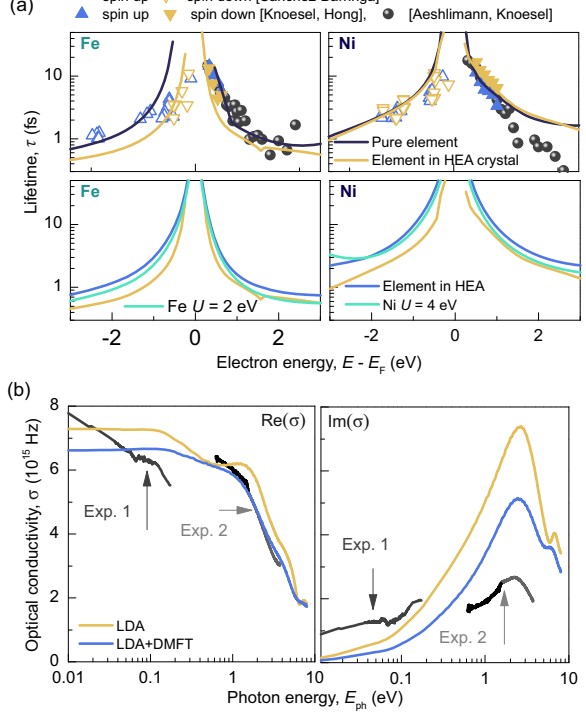

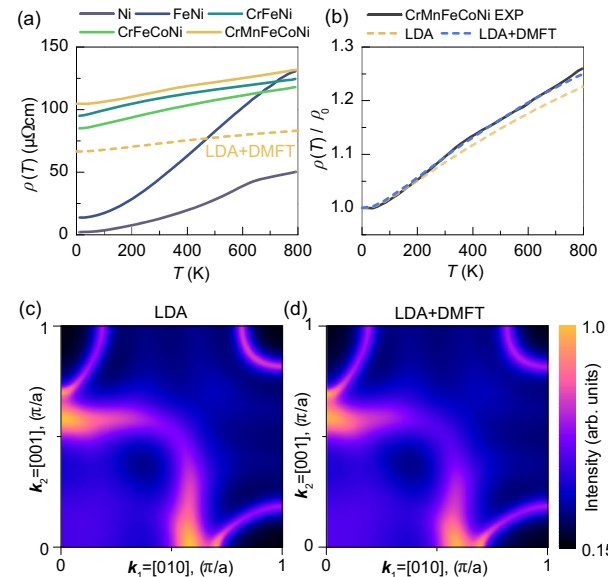

**Fig. 3 | Element specific quasiparticle lifetimes ($\tau$) and frequency resolved optical conductivity of the CrMnFeCoNi HEA. a** $\tau$ from LDA + DMFT calculations for Fe and Ni. Top: Black lines show calculated $\tau$ for pure elements in their natural crystal structures; yellow lines show $\tau$ for pure metals within the fcc structure and CrMnFeCoNi lattice constant ($\gamma$-Fe). Experimental values are from photoemission spectroscopy (PES, below $E_F$)[43] and time-resolved two-photon photoemission (TR-2PPE, above $E_F$)[44–46] of pure elements. Ni shows negligible variation, while $\gamma$-Fe exhibits notably reduced lifetimes. Bottom: Blue lines correspond to the CrMnFe-CoNi HEA results. Green lines show results with increased $U$ for Fe and Ni (2 eV and 4 eV, respectively). Increasing $U$ reduces $\tau$, particularly for Ni at $E > E_F$. **b** Real (Re($\sigma$)) and imaginary (Im($\sigma$)) parts of the complex optical conductivity ($\sigma$) versus photon energy ($\hbar\omega$). LDA calculations are shown as yellow lines, and LDA + DMFT calculations as blue lines. Experimental data from reflectometry (Exp. 1) and ellipsometry (Exp. 2) are given by black lines and marked by arrows. LDA + DMFT results show better agreement with experimental data, especially in the visible and UV ranges, for both Re($\sigma$) and Im($\sigma$).

**Fig. 4 | Temperature dependent electrical resistivity and fermi surface.**
**a** Electrical resistivity ($\rho$) measurements vs. temperature ($T$) for Ni, FeNi, CrFeNi, CrFeCoNi, and CrMnFeCoNi. The ternary to quinary alloys show a sharp increase in residual resistivity, while Ni and FeNi exhibit quasiparticle temperature dependency. The LDA + DMFT calculation for CrMnFeCoNi is shown by the yellow dashed line. (**b**) Comparison of experimental resistivity for CrMnFeCoNi (black line) with LDA (yellow dashed line) and LDA + DMFT (blue line) calculations, normalized by their residual resistivity. The LDA + DMFT results show better agreement with experimental data, particularly at higher temperatures, where states farther from the Fermi edge contribute to transport. (**c** and **d**) Fermi surface of CrMnFeCoNi calculated using LDA (**c**) and LDA + DMFT (**d**). Color maps represent spectral intensity. Both graphs show smeared Fermi surfaces, with chemical disorder significantly affecting states near $E_F$. Many-body effects do not introduce additional smearing.

resonance position in Im($\sigma(\omega)$) is found at $\hbar\omega$ = 2.3 eV experimentally, and 2.7 eV in the LDA and at 2.4 eV in the LDA + DMFT calculations. Besides improving spectral position, LDA + DMFT calculations reduce the resonance maximum significantly, achieving an improved experimental alignment.

## Temperature dependence of electrical resistivity

Revisiting temperature-dependent electrical resistivity, we focus on correlation effects. Figure 4a depicts our four-point resistivity measurements for different metals, from Ni to CrMnFeCoNi HEA, over a broad temperature range (4 K to 800 K). A dominant increase in residual resistivity is observed when transitioning from FeNi to CrFeNi, which is attributed to the coupling of ferromagnetic and antiferromagnetic elements within the alloys[12]. This results in smearing across both spin channels in the BSF of CrFeNi, whereas FeNi exhibits solely smearing in the minority spin channel[12]. Thus, Ni and NiFe show well-defined quasiparticle transport properties within the majority channel with progressive increase in upward curvature up to Curie temperature at 625 K and to 835 K, respectively. For the alloys exhibiting higher residual resistivity, similar trends are observed: $\rho_0$ ranges around 100 µΩcm, and d$\rho$/d$T$ remains low (and after subtraction of $\rho_0$, d$\rho$/d$T$ seems to be identical).

We compare the CrMnFeCoNi measurement with linear response calculation within the Kubo-Greenwood formalism[55]. The alloy analogy model[55] is employed to mimic temperature-dependent lattice vibrations, inherently integrating electron-phonon scattering processes[56]. Corroborating the BSF calculations, both LDA and LDA + DMFT results yield comparable $\rho_0$ of 67 µΩcm. By Fermi surface analysis the

energy, and thus scattering rates, when considering open $d$-band metals.

Calculation results are depicted in Fig. 3b (LDA in yellow, LDA + DMFT in blue) alongside experimental data (see SI). The left subfigure displays the real part of the optical conductivity, Re($\sigma(\omega)$), corresponding to the absorptive component. The results from LDA and LDA + DMFT show for low $\hbar\omega$ a constant trend with a minor offset between each other, and dispersive variations in the VIS range where both curves decline. Quasiparticle lifetimes from the DMFT scheme improve the absorptive part of $\sigma(\omega)$ drastically, especially in the visible and UV range, where a perfect agreement with experiment is found. Here the pure LDA calculations underestimate the absorption. Also for the LDA + DMFT case, at photon energies of 5 eV and above, the fine structure found in the LDA calculation is blurred (see small oscillation between 6 eV and 8 eV for real and imaginary part). This behavior reflects the smearing of sub-bands, induced through the quasi-particle lifetimes, as seen within the BSF in Fig. 3c, d. For Im($\sigma(\omega)$), corresponding to the dispersive or reflective part, the situation is similar. The experiments show low values for small $\hbar\omega$ (approaching zero), comparable to the Drude-behavior, with a resonance peak in the VIS region. The LDA calculation overestimates this peak threefold and is non-symmetric on a log scale, thus not purely Drude-like. The

resistivity may be calculated from the $k$-space smearing (electron mean free path), as well the surface area[8]. Figure 4c, d show the Fermi surfaces calculated with the LDA and LDA + DMFT potentials, which exhibit remarking similarities. This supports our linear response results. However, the substantial deviation from the experimental value of 105 μΩcm could be influenced by potential short-range ordering[57–59], which has already been experimentally shown to increase resistivity[60]. Such mechanisms are not included within the CPA framework and need to be captured with more sophisticated methodologies like non-local CPA[61,62]. The localization mechanism must also be considered. While Hubbard localization is addressed within the LDA + DMFT framework, the onset of Anderson localization is inherently not captured by the CPA approach[63,64]. The temperature dependence of $\rho(T)$ is presented in a normalized format $\rho(T)/\rho_0$ in Fig. 4b and demonstrates that while LDA closely aligns with experimental data, even indicating a reduced d$\rho$/d$T$, LDA + DMFT yields perfect agreement across all temperatures. This aligns with the BSFs, where LDA + DMFT exhibits increased smearing at higher $E_B$, thus impacting electronic transport at elevated temperatures. For a direct comparison of the LDA + DMFT calculation with the experiments, the results are presented also in Fig. 4a as dashed yellow line. A comparison clearly shows that the temperature dependence is well-represented, despite the already discussed offset in the residual resistivity.

Our findings identify CrMnFeCoNi as a material characterized by pronounced electronic correlations. These correlations exist in addition to the prevailing chemical and magnetic disorder, which smear the band structure in the vicinity of $E_F$ and thus primarily cause the high residual resistivity. The extent of many body effects by means of on-site Coulomb interaction within the Hubbard model mirrors those of the containing pure elements. Correlation effects gain in significance with increasing distance from the Fermi edge, which was demonstrated both experimentally and theoretically utilizing electronic spectroscopies and temperature dependent electronic transport. Especially in the calculation of the optical response, accounting for quasiparticle lifetimes dramatically improve the absorptive as well dispersive part of the optical conductivity in the VIS-UV range, as energy dependent electron-electron scattering may overrule electron-phonon scattering rates.

Our findings broaden the understanding of electronic correlations in CrMnFeCoNi HEA, offering a robust framework for exploring the complex electronic structures of various CCAs. This deeper insight enables a more accurate prediction and optimization of electronic structure-derived properties, such as thermophysical, transport, and optical properties, which is vital for the future development of advanced materials for applications in medicine, industry, and science.

## Methods
### Sample preparation and characterization
All alloys were first synthesized by mixing and pressing powders of elemental metals (total mass 1.5 g) into pellets to achieve the targeted composition. The pellets were placed in an arc-melting chamber, with glowing elemental zirconium used to remove any residual oxygen. Arc melting was performed three times, and the alloys were subsequently annealed in an evacuated quartz ampoule for one month at 1030 °C to enhance sample homogeneity. The specimens were cut into approximately 0.5 mm thick disks for photoemission and optical conductivity measurements and into bars of dimensions 8 × 3 × 0.5 mm³ for four-point electrical resistivity measurement. The series of prepared samples included Ni, NiFe, NiFeCr, NiFeCrCo, and NiFeCrCoMn. Characterization of the HEA samples using X-ray diffraction confirmed a well-defined fcc crystalline structure. Scanning electron microscopy and electron-dispersive X-ray spectroscopy analysis revealed the actual composition and satisfactory dispersion of constituent elements.

### Electronic spectroscopies measurements
XAS and ResPES measurements were conducted at the soft-X-ray ARPES endstation at the ADRESS beamline[65,66] of the Swiss Light Source, Paul Scherrer Institute, Switzerland. The CrMnFeCoNi alloy was kept at a base temperature of 20 K. The photon energy was scanned along each of the relevant $L_3$ absorption edges at steps of 100 meV, while at the same time recording a valence band spectrum with a hemispherical electron analyzer at a resolution better than 100 meV.

### Optical conductivity measurements
For the optical conductivity measurements, the CrMnFeCoNi sample was ground with a SiC paper and subsequently polished with diamond paste with decreasing grain sizes to achieve a root mean square surface roughness of 2 nm. The spectra were obtained by ellipsometry (Sentech SE 850) in the visible to infrared range and by reflectance measurements with subsequent Kramers-Kronig transformation in the far infrared part of the elecromagnetic spectrum.

### LDA + DMFT calculations
DFT calculations of CrMnFeCoNi were performed within the fully relativistic spin polarized multiple scattering Korringa-Kohn-Rostoker (SPR-KKR) Greens function formalism[67,68]. Many-body correlation effects beyond local density approximation (LDA) were added via DMFT, as implemented in SPR-KKR[52]. Chemical disorder was accounted for by the coherent potential approximation (CPA)[69,70] and the paramagnetic state above 20 K[71] was mimicked by the disordered local moment scheme[72]. An appealing feature of our approach is that it allows to consider local quantum and disorder fluctuations on the same footing. This has also already been successfully realized by other groups[73]. For LDA + DMFT calculations, the Hubbard U for 3d electrons was set to the values found for pure elements, while the Hund exchange was J = 0.94 eV for all elements. The U values for Cr, Mn, Fe, Co, Ni were equal to 2.0 eV[74], 3.0 eV[41], 1.5 eV, 2.5 eV and 3.0 eV[43], respectively. A variation in J was not considered significant, as it has only marginal effects on the electronic structure and related spectra within the choosen calculation scheme. For further details, see SI.

### Application of the Cini-Sawatzky theory (CST)
We applied a self-convolution to the calculated partial density of states and compared the results within the CST[32–34] with the experimental data obtained fromResPES. This approach aims to replicate the two-hole interactions using single-particle ground states, necessitating the presentation of the resultant self-convolution on a two-particle energy scale[37]. Consequently, the new energy axis is multiplied by a factor of two. Prior to convolution, the pDOS data were interpolated on a refined energy grid.

### Calculation of spectroscopic and transport properties
In our study, valence band photoemission[42] and electronic transport calculations, including the optical conductivity tensor, were conducted using the SPR-KKR code and a Kubo-framework for linear response, respectively. These calculations incorporated electron-phonon interactions within the alloy analogy model[55], considering temperature-dependent atomic displacements and a comprehensive k-point mesh for accuracy. The optical response to external electromagnetic fields was detailed through a current-current correlation function reformulated in Green's function terms, capturing the Drude contribution within a fully relativistic framework[49]. Our findings primarily focus on the isotropic $\sigma(\omega)$ in paramagnetic CrMnFeCoNi HEA. For an in-depth explanation of methodologies, including the computational parameters and models employed, the reader is reffered to the SI.

## Data availability

The data generated and presented in this study are available in the main article and the Supplementary information. Source data are provided as a Source Data file. Source data are provided with this paper.

## Code availability

The SPRKKR multiple scattering package is freely available (no costs apply) under the specific user license and the package can be downloaded following registration at https://www.ebert.cup.uni-muenchen.de/index.php/en/software-en. All input files, post-processing procedures and scripts used to evaluate experimental and theoretical data are available from the authors on request.

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

## Acknowledgements

This work was supported by Deutsche Forschungsgemeinschaft under Grant 528706678 awarded to H.H. and the project MEBIOSYS, funded as project No. CZ.02.01.01/00/22_008/0004634 by Programme Johannes Amos Commenius, call Excellent Research awarded to J.M. This research is also part of the project No. 2022/45/P/ST3/04247 co-funded by the National Science Center and the European Union's Horizon 2020 reseach and innovation program under the Marie Skodowksa-Curie grant agreement no 945339, awarded to I.D.M. For the purpose of Open Access, the author has applied a CC-BY public copyright licence to any Author Accepted Manuscript (AAM) version arising from this submission.

## Author contributions

J.M., L.F., and H.D. developed the concept. E.M., X.M., L.C., D.T., and T.I. prepared the samples and performed transport measurements. M.C., E.B.G., V.S., and H.D. were responsible for the photoemission experiments at the Swiss Light Source. D.R. and J.M. conducted the CST analysis. D.R. and S.K. carried out the DFT calculations. I.D.M. interpreted the DMFT results, and H.H. discussed the optics. H.E., J.M., and A.H. developed the KKR package. D.R. wrote the manuscript in close collaboration with J.M., L.F., and H.D. All authors contributed to discussions and commented on the manuscript.

## Competing interests

The authors declare no competing interests.
