## [Peer Review File · Nature Communications]

Interplay between disorder and electronic correlations in compositionally complex alloysREVIEWER COMMENTS

Reviewer #1 (Remarks to the Author):

The paper describes a highly impressive and thorough investigation of properties of the high entropy Cantor-Wu alloy, CrMnFeCoNi, via photoemission spectroscopy, optical conductivity and electrical resistivity experiments, all of which are analysed with detailed, quantitative theoretical calculations. The work is a fine case study of the rich links between electron correlations, nature of self energy effects and multicomponent disorder.

Element-specific information extracted from resonant PES and X-ray absorption data show the effects on these data of localised electron correlations in concert with chemical disorder and magnetic fluctuations. Electron correlation effects are modelled effectively by DFT+DMFT within a KKR-CPA framework using prescribed U values for local onsite correlations. Element-specific information such as quasi-particle lifetimes is also extracted from optical conductivity measurements and appropriate rigorous theoretical analysis. Finally the authors show that localised correlation effects appear to have less effect on the temperature-dependent electrical resistivity calculations which undershoot the experimental measured values. They identify important aspects such as atomic short-range order and Anderson-type localisation effects that require further study.

The paper is very interesting and suitable for publication in Nature Communications. The manuscript, however, could be improved by some further proof-reading and also making the figure captions more accessible and easier to follow - the reader could be reminded about what some of the acronyms stand for (e.g. CST, TR-2PPE) and directed to note certain features.

Reviewer #2 (Remarks to the Author):

This manuscript delves into the exploration of disorder and electronic correlations within the CrMnFeCoNi high-entropy alloy using photoemission spectroscopy, and the results are

compared with DFT/DMFT calculations. This is a timely piece of work because detailed investigations into the electronic structure of these alloys, particularly with regard to the elemental contributions to the valence band structure, is still very rare. While the quality of the work is commendable, this reviewer suggests that the manuscript may find a better fit in a specialized journal with a stronger physics focus. The findings, while of importance, may lack the broader impact required for publication in a top-tier journal like this.

Reviewer #3 (Remarks to the Author):

In this manuscript, the authors investigated the electron correlation effect on the band structure and transport properties in HEA-CrMnFeCoNi using both experimental and theoretical methods. The authors found that the KKR-CPA+DMFT method explains the experimental results well, showing the importance of electron correlation even in HEAs. The authors also discuss the differences between LDA and LDA+DMFT in detail by comparing low-energy and high-energy electronic transport properties, emphasizing the importance of strong correlation effects, especially in the high-energy regime. This study advances the understanding of electronic states in HEAs and provides significant information to those working in this field.

I have the following comments regarding this manuscript:

(1) In the Methods section, while the element dependence of U is discussed in detail, the Hund coupling J is treated as element-independent. The authors should explain the reason for this.

(2) The effect of J on the electronic states should be discussed.

(3) In Fig. 4(a), the LDA+DMFT calculation results should also be included.

Based on the above, I think that this manuscript is suitable for publication in Nature Communications after addressing my concerns.

Response to Reviewer's remarks on the manuscript entitled "Interplay between disorder and electronic correlations in compositionally complex alloys" (NCOMMS-24-11440)

We sincerely appreciate the valuable feedback and constructive comments provided by the Reviewers. The consistently positive feedback on the quality and results of our work is very pleasing. The manuscript has been revised to address all of the concerns and suggestions raised. Below is a summary of the changes made, followed by specific responses to each Reviewer.

All changes to the manuscript and Supplementary Information are highlighted: new text in blue color, and old, removed text in red color and crossed out.

Main Manuscript:

- The abstract has been modified to be more appealing and to better generalize the scope of the work.
- The final section of the introduction has been revised to more effectively guide the reader through the paper and present the generalizable results.
- Two sentences have been appended to the conclusion in order to enhance its efficacy.
- In order to facilitate the reading of the manuscript regarding the ResPES analysis for the reader, the corresponding section has been optimized. In addition to a few minor changes, including a reference to a new section in the supplement, some sentences have been added to clarify the theoretical background.
- In order to address the issue raised by Reviewer 3, a sentence redirecting the reader to the Supplemental Information for the analysis of the role played the Hund exchange J has been added to the Methods section.
- The x label in Figure 1 has been changed from BE to EB to correct a minor inconsistency.
- Figure 2 has been reformatted from a single-column layout to a two-column layout to improve the visibility of the spectra displayed, thus improving accessibility. The subfigures have not been changed.
- LDA+DMFT results for electrical resistivity have been added to Figure 4(a) as suggested by Reviewer 3. An appropriate sentence has been added to the main text.
- All figure captions have been revised to improve clarity and accessibility, as requested by Reviewer 1.
- Thorough proofreading was performed to correct inconsistencies and irregularities.
- Eduardo Bonini Guedes has been added as a co-author, as he was overlooked in the initial submission. He was involved in the ResPES measurements. His contribution has been updated in "Author Contributions". He has been given the latest version of the manuscript to review, like the other authors.
- The current address of Trpimir Ivšić has been added.
- A code availability statement has been added.

Supplementary Information:

- A new section (1) has been added to provide more detailed plots on the ResPES data.
- A new section (3) has been added to discuss the influence of J in our calculations, including relevant new references.
- All figures and sections were renumbered to accommodate the new sections.
- All figure captions have been refined at the request of Reviewer 1.
- Eduardo Bonini Guedes has been added as a co-author (see above).
- The current address of Trpimir Ivšić has been added.

Reviewer #1 (Remarks to the Author):

The paper describes a highly impressive and thorough investigation of properties of the high entropy Cantor-Wu alloy, CrMnFeCoNi, via photoemission spectroscopy, optical conductivity and electrical resistivity experiments, all of which are analysed with detailed, quantitative theoretical calculations. The work is a fine case study of the rich links between electron correlations, nature of self energy effects and multicomponent disorder.

Element-specific information extracted from resonant PES and X-ray absorption data show the effects on these data of localised electron correlations in concert with chemical disorder and magnetic fluctuations. Electron correlation effects are modelled effectively by DFT+DMFT within a KKR-CPA framework using prescribed U values for local onsite correlations. Element-specific information such as quasi-particle lifetimes is also extracted from optical conductivity measurements and appropriate rigorous theoretical analysis. Finally the authors show that localised correlation effects appear to have less effect on the temperature-dependent electrical resistivity calculations which undershoot the experimental measured values. They identify important aspects such as atomic short-range order and Anderson-type localisation effects that require further study.

The paper is very interesting and suitable for publication in Nature Communications. The manuscript, however, could be improved by some further proof-reading and also making the figure captions more accessible and easier to follow - the reader could be reminded about what some of the acronyms stand for (e.g. CST, TR-2PPE) and directed to note certain features.

Authors:

We sincerely appreciate Reviewer 1's detailed and positive feedback on our manuscript. We have carefully addressed all the points raised, including proof-reading and enhancing the figure captions for clarity and accessibility. We defined all abbreviations in the captions, pointed out important features and added physical explanations (if they are not too long). We believe these revisions have significantly improved the quality of the manuscript.

Reviewer #2 (Remarks to the Author):

This manuscript delves into the exploration of disorder and electronic correlations within the CrMnFeCoNi high-entropy alloy using photoemission spectroscopy, and the results are compared with DFT/DMFT calculations. This is a timely piece of work because detailed investigations into the electronic structure of these alloys, particularly with regard to the elemental contributions to the valence band structure, is still very rare. While the quality of the work is commendable, this reviewer suggests that the manuscript may find a better fit in a specialized journal with a stronger physics focus. The findings, while of importance, may lack the broader impact required for publication in a top-tier journal like this.

Authors:

We would like to thank Reviewer 2 for the invested time and positive scientific evaluation of our work. However, we respectfully disagree with the assertion that our manuscript "may lack the broader impact required for publication in a top-tier journal like this." We believe that our study is highly suitable for Nature Communications, and we would like to highlight a few points:

We think that our study represents a significant advancement as it provides, to our knowledge, the first quantitative comparison of the influence of disorder and correlated many-body effects on electronic structure-driven properties in CCAs/HEAs. Based on the results, we are able to make clear statements regarding electronic spectra, transport and optics. In principle, these can be generalized to other multi-principal element alloys and are therefore suitable for many other technically important advanced materials.

Building on recent findings on CCAs, such as studies on short-range order [Nature 624, 564-569 (2023); Nat. Commun, 13, 1021 (2022); Nat. Commun. 581, 283-287 (2020)], or nano-structure induced mechanical properties [Nat. Commun. 15, 4599 (2024), Nat. Commun. 14, 2516 (2023)], we describe an additional physical mechanism that influences key material properties. Further, we want to highlight that some papers were recently published in Nature Communications regarding electronic and phononic transport [Nat. Commun. 15, 4554 (2024), Nat. Commun. 13, 7509 (2022)], utilizing specific HEAs as model materials and providing generalizable results. We would also like to emphasize the potential of the results to extend DFT-based machine learning models [Science 378, 6615, 78-85 (2022)] to account for many-body correlation effects. From our perspective, this appears to be crucial for the further development of alloys with improved transport and optical properties.

Regarding optical properties, our findings are also interesting for the photonics industry, which operates largely in the visible and near-infrared spectral regions. Alloys are of decisive importance for these applications, especially in laser processing, where the initial absorption of laser radiation is the central part of the process. Here we can clearly demonstrate that through the implementation of correlated many-body effects, the predicted optical spectra are improved.

Another point is that the KKR-CPA approach is extremely valuable for disordered alloys and is attracting increasing interest from the materials science community, especially for HEA calculations. We also wish to emphasize the extremely positive feedback from the other two Reviewers, including their suggestion for publication.

Finally, we understand that the broader implications of our study were not immediately apparent in the previous version of our manuscript, due to the specific and concise nature of our abstract and conclusion. Therefore, we have added some sentences at the beginning of the summary and at the end of the conclusion to attract the interest of a wider audience.

We hope these revisions and clarifications will convince the Reviewer of the broader impact and significance of our work.

Reviewer #3 (Remarks to the Author):

Reviewer:

In this manuscript, the authors investigated the electron correlation effect on the band structure and transport properties in HEA-CrMnFeCoNi using both experimental and theoretical methods. The authors found that the KKR-CPA+DMFT method explains the experimental results well, showing the importance of electron correlation even in HEAs. The authors also discuss the differences between LDA and LDA+DMFT in detail by comparing low-energy and high-energy electronic transport properties, emphasizing the importance of strong correlation effects, especially in the high-energy regime. This study advances the understanding of electronic states in HEAs and provides significant information to those working in this field.

I have the following comments regarding this manuscript:

Authors:

The authors would like to thank Reviewer 3 for his time and valuable comments. We are also very pleased to receive positive feedback on our work. We have addressed all the points raised and are confident that our revisions have improved the quality of the manuscript.

Reviewer:

(1) In the Methods section, while the element dependence of U is discussed in detail, the Hund coupling J is treated as element-independent. The authors should explain the reason for this.

Authors:

In principle, we could have introduced a small variation of the Hund exchange to mimic what has been reported for the $3d$ elemental series. Calculations based on constrained random-phase approximation (cRPA) show that the Hund exchange changes of about 0.1 eV when going from Cr to Ni [Phys. Rev. B **77**, 085122 (2008)]. This variation may be 50% larger in our case, accounting for differences in the local orbitals [see also Ref. 49 in the manuscript]. There are, however, two main reasons why this variation is not fully meaningful in our case. First, our study is focused on a high-entropy alloy and not on pure elements. Cr, Fe and Co have a different crystal structure than the one investigated here, with a different number of neighbors and a different screening. Second, a change of J of about 0.1 or 0.2 eV would have no visible

effect on our calculations. This is due to the particular computational scheme we use for our study, where the DMFT self-energy acts as a correction to a spin-polarized DFT solution. In this approach, which is the most common way to apply DFT+DMFT to magnetic materials, the major effect associated to J , i.e. the renormalization of the exchange splitting, is completely canceled by the double-counting term [see e.g. Phys. Rev. B **97**, 184404 (2018) for the analysis of the consequences on the effective magnetic coupling]. A detailed investigation of these methodological subtleties has been provided for DFT+U, in Phys. Rev. B **98**, 125126 (2018). On the left, we show Fig. 2 of this work, focused on

LaMnO₃. In panel (a), various exchange splittings are defined. In panel (b), these values are plotted with respect to the variation of the Hund exchange J . Variations of the order of 0.1 or 0.2 eV cause minor changes in the splitting and thus in the spectrum. These changes are due to the fact that J also determines the anisotropic effects among the different orbitals of the 3d shell. To restore a more significant dependence, one should work with a non-spin-polarized DFT solution and let the whole magnetism arise from the local problem in DMFT or DFT+U. This is illustrated by the data plotted in panel (c). Although these data are for DFT+U, similar trends are expected for DFT+DMFT. The presence of higher order terms beyond Hartree-Fock is likely to be compensated by the reduction of the effective interaction due to dynamical screening [see e.g. discussions in Phys. Rev. B **90**, 165130 (2014)].

Reviewer:

(2) The effect of J on the electronic states should be discussed.

Authors:

In the previous answer, we have explained why a small variation of J is expected to have a negligible effect on the final spectra. We now aim to demonstrate this, as suggested by the referee, on systems related to our study. For this purpose, we calculated the influence of a J variation on the electronic spectra for ferromagnetic (FM) Ni and paramagnetic (DLM) FeNi, using the SPRKKR package. J is changed at both Fe and Ni sites, while the U values are kept to the same values as those reported in the paper. The density of states (DOS) resulting from these calculations is shown in the following figure, panels (a) to (c).

Even a variation of J as large as ± 0.4 eV (from $J = 0.5$ eV to 1.3 eV) does not lead to a significant shift in the spectral weight of the d-bands for FM Ni (see excitation energies between -5 eV and 0 eV). The most visible effect is a slight shift of the split-off satellites. For ferromagnetic Ni, majority (up) channel, the satellite shifts from -8.2 eV to -8.7 eV as J increases from 0.5 eV to 1.3 eV. This corresponds to a shift of $\Delta P_s / \Delta J = -0.6$ (P_s is the position of the spectral weight of the satellites). For the minority (down) spin channel, the satellite is less marked and appears as a shoulder-like feature between -5 and -10 eV. Increasing J causes a

farther attenuation. For paramagnetic FeNi, even smaller changes in the electronic spectra are observed for J variations from 0.6 eV to 1.2 eV. As for the minority spin channel of FM Ni, a slight influence on the shoulders around -8 eV is found. No changes in the bandwidth or shift of the d -block are observed upon a variation of J .

Finally, in panel (d), we show how the results of the calculations for paramagnetic (DLM) CrMnFeCoNi depend on varying J . The U values again correspond to those in the main manuscript. Unlike FeNi, we find no influence on the shoulder features at -8 eV, which means that the simultaneous variation of J at the various sites leads to compensating effects. The main peak position remains unchanged at -1 eV for all J , while its height only slightly decreases from 2.01 states/eV/atom to 1.97 when J goes from the lowest to the largest value. The DOS at E_F is completely unaffected. The spectral width of the d -band block does not change, but the spectral weight shifts slightly towards higher binding energies with increasing J . This is observed in the flanks of the d -band block, which shift approximately with $\Delta P_f/\Delta J = -0.3$ (P_f is the position of the d -band flanks). Compared to the effect induced by a variation of U , as shown in Figure 2(a) of the main manuscript or Figure 3(b) in the supplementary materials, the uncertainty of J is negligible. An increase of U for Ni from 3 eV to 4 eV in CrMnFeCoNi induces a direct and isolated shift of the Ni satellite from -8 eV to -10 eV. This corresponds to $\Delta P_s/\Delta U = -2$. Considering that the expected variation of J is of 0.1 eV to 0.2 eV, as discussed in the previous question, we can safely ignore this effect.

We agree with the referee that these issues should be addressed in our manuscript. Therefore, we have summarized the main points of the previous discussions in the Supplementary Information, where we also included the previous results. In the main manuscript, we added a reminder to this discussion in the Methods section, just after mentioning our choice of keeping the Hund exchange constant for all the elements considered.

Reviewer:

(3) In Fig. 4(a), the LDA+DMFT calculation results should also be included.

Authors:

Figure 4(a) shows the experiments on the electrical resistivity of various alloys, ranging from Ni to CrMnFeCoNi. Due to the lack of short-range order in our CPA-based calculations, we observed a larger offset between measurements and calculations. This issue has already been discussed in the manuscript. In response to the referee's suggestion, we have now included the LDA+DMFT calculations (see dashed line in the revised manuscript).

REVIEWERS' COMMENTS

Reviewer #1 (Remarks to the Author):

The authors have satisfactorily addressed points raised in my first report and have improved the manuscript. The paper is suitable for publication in Nature Communications.

Reviewer #2 (Remarks to the Author):

Like I mentioned in my previous review, the manuscript is of good quality and provides important missing data in the current understanding of CCAs. My concern about this manuscript lies in its limited broader impact. More specifically, the new data on the electronic structure of this model alloy does not seem to bring significant new insights or enhance our general knowledge of how CCAs differ from conventional alloys. The authors attempt to address this in their revised manuscript, but the connection between the new data and new insights remains unclear to me. However, as scientific importance is somewhat subjective, I will leave the final decision to the editor.

Reviewer #3 (Remarks to the Author):

The authors have provided a satisfactory response to my comments. Therefore, I have no further concerns, and I believe that this manuscript is suitable for publication in Nature Communications.

Response to Reviewer's remarks on the manuscript entitled "Interplay between disorder and electronic correlations in compositionally complex alloys" (NCOMMS-24-11440)

We sincerely thank the reviewers for their thorough evaluation and positive feedback on our revised manuscript. The constructive comments have undoubtedly improved the manuscript and we are pleased that it is now suitable for publication in Nature Communications.

As the reviewers did not raise further questions or suggest any additional changes, we have not revised the manuscript further.

We greatly appreciate your time and effort in reviewing our work.

On behalf of all coauthors,

Ján Minár